# ZF-AutoML: An Easy Machine-Learning-Based Method to Detect Anomalies in Fluorescent-Labelled Zebrafish

**Ryota Sawaki [1,†]** , **Daisuke Sato [2,†]** , **Hiroko Nakayama [3,4]** , **Yuki Nakagawa [2]** and **Yasuhito Shimada [1,4,5,*]**

1   Department of Integrative Pharmacology, Mie University Graduate School of Medicine, Tsu, Mie 514-8507, Japan; 316052@m.mie-u.ac.jp
2   RT Corporation, Chiyoda ku, Tokyo 101-0021, Japan; tiryoh@gmail.com (D.S.); nakagawa@rt-net.jp (Y.N.)
3   Graduate School of Regional Innovation Studies, Mie University, Tsu, Mie 514-8507, Japan; 27293301@m.mie-u.ac.jp
4   Mie University Zebrafish Drug Screening Center, Tsu, Mie 514-8507, Japan
5   Department of Bioinformatics, Mie University Advanced Science Research Promotion Center, Tsu, Mie 514-8507, Japan
*   Correspondence: shimada.yasuhito@mie-u.ac.jp; Tel.: +81-59-231-5384
†   These authors contributed equally to this work.

**Abstract:** Background: Zebrafish are efficient animal models for conducting whole organism drug testing and toxicological evaluation of chemicals. They are frequently used for high-throughput screening owing to their high fecundity. Peripheral experimental equipment and analytical software are required for zebrafish screening, which need to be further developed. Machine learning has emerged as a powerful tool for large-scale image analysis and has been applied in zebrafish research as well. However, its use by individual researchers is restricted due to the cost and the procedure of machine learning for specific research purposes. Methods: We developed a simple and easy method for zebrafish image analysis, particularly fluorescent labelled ones, using the free machine learning program Google AutoML. We performed machine learning using vascular- and macrophage-Enhanced Green Fluorescent Protein (EGFP) fishes under normal and abnormal conditions (treated with anti-angiogenesis drugs or by wounding the caudal fin). Then, we tested the system using a new set of zebrafish images. Results: While machine learning can detect abnormalities in the fish in both strains with more than 95% accuracy, the learning procedure needs image pre-processing for the images of the macrophage-EGFP fishes. In addition, we developed a batch uploading software, ZF-ImageR, for Windows (.exe) and MacOS (.app) to enable high-throughput analysis using AutoML. Conclusions: We established a protocol to utilize conventional machine learning platforms for analyzing zebrafish phenotypes, which enables fluorescence-based, phenotype-driven zebrafish screening.

**Keywords:** artificial intelligence; fluorophores; in vivo screening

## 1. Introduction

Zebrafish is considered one of the main animal models for in vivo chemical testing and toxicity evaluation. Due to their transparent body, phenotypic evaluation conducted for several screening experiments, also known as "zebrafish screening", can be done using live imaging of the target organs by dye staining or by introducing transgenes for fluorescent proteins. To evaluate the phenotypic changes, several types of image analysis procedures (software program, macros, and pipelines) have been developed and applied for large-scale zebrafish screening, such as ImageJ [1], custom R scripts [2],

or commercial ones [3,4]. In general, freeware programs developed by researchers are semi-automatic and sometimes challenging to use, while commercial software programs for high-content images are easy to use, fully automatic, but expensive. We also developed a simple and free fluorescence quantification software for processing multiple zebrafish images (available on Windows and macOS) that can be used for zebrafish screening [5]. However, each zebrafish image still needs to be visually observed by skilled researchers to ensure that the phenotypic changes are properly identified during the experiments.

Machine learning is an algorithm based on statistical models that are used by computer systems to perform a specific task effectively by relying on patterns and inference and that do not require explicit instructions. Machine learning has been applied broadly in many fields, such as image recognition. In 2005, machine learning was applied for the first time in zebrafish research to analyze zebrafish transcriptomics as support vector machines [6]. Further, machine learning has been used for behavior [7,8] and image analysis [9–11]. Cordero-Maldonado et al. recently reported that a combination of machine learning and robotic manipulation enables microinjection into zebrafish eggs at a trained target site in a high-throughput manner [12]. The combination of automatic imaging, quantification, and machine learning should promote large-scale, high-throughput zebrafish screening in the field of drug discovery. The attractiveness of machine learning is its superior efficacy, although the analysis pipeline is usually specific for a single purpose and cannot be easily generated, requiring computer specialists and having a high cost. That means small-to-medium labs rarely use these systems.

In this study, we developed an easy analysis pipeline for zebrafish, particularly the fluorescently labelled ones, using Google AutoML. AutoML is a web-based application used for creating machine learning-based methodologies and it is free for an upload of up to 3000 images. We conducted 2 types of zebrafish experiments using vascular- and macrophage-specific Enhanced Green Fluorescent Protein (EGFP) expression strains and showed that AutoML can be applied for zebrafish screening with a few modifications.

## 2. Materials and Methods

### 2.1. Ethic Approval

All animal procedures were performed according to the Japanese animal welfare regulatory practice Act on Welfare and Management of Animals (Ministry of Environment of Japan) in compliance with international guidelines. Ethical approval from the local Institutional Animal Care and Use Committee was not sought, as this law does not mandate the protection of fish.

### 2.2. Zebrafish Experiments

The zebrafish *Tg (kdrl:EGFP)* strain was a kind gift by Prof. Stefan Schulte-Merker [13], and the *Tg (mpeg1:EGFP)* strain was purchased from the Zebrafish International Resource Center (Eugene, OR, USA). The zebrafish were maintained in our facility according to standard operational guidelines. For the angiogenesis assay, *Tg (kdrl:EGFP)* zebrafish were treated with sorafenib (SignalChem, Richmond, BC, Canada) from 24 h post fertilization (hpf) to 96 hpf, according to the method reported in a previous study [14]. For the macrophage assay, we used *Tg (mpeg1:EGFP)* zebrafish [15]. The caudal fin of the 72 hpf zebrafish were injured using a fine glass blade and the images of the macrophages were taken 6 h after the injury.

### 2.3. Image Capture

The zebrafish images were captured using an all-in-one fluorescence microscope (BZ-X710; Keyence, Osaka, Japan) equipped with a Nikon CFI 60 Series infinite optical system and 2.83 million-pixel monochrome charged coupled device (CCD [output signal is 14 bit]) camera. For imaging, zebrafish were anaesthetized with 0.003% tricaine (MS222; Sigma-Aldrich, St. Louis, MO,

USA) and mounted laterally on 10-cm dishes coated with 1% agarose dissolved in $H_2O$. The conditions for image capture were as follows:

4× objective lens with 10× eyepieces (40× total magnification).
High-resolution mode (1920 × 1440 px).
Exposure time; brightfield: 1/7500 s, green fluorescent protein
(GFP: Ex 470/40, Em 525/50): 1.2 s.
The orientation of zebrafish was random.

Merged images (automatically prepared using the BZ-H3AE software; Keyence) were used for this study.

### 2.4. Image Processing

Images of the macrophage assay were pre-processed for machine learning. Microsoft Paint was used to partially cut the image including the wound area (size: 631 × 601 px). ImageMagick (Version 7.0; ImageMagick Studio, Landenberg, PA, USA) was used to rotate the images. ImageMagick and GIMP software (version 2.10; C Orinda, CA, USA) were used to emphasize the green color of the images. For a step-by-step protocol, please see the Supplementary Method. In addition, to reduce the file size and upload time of images to AutoML, we converted TIFF files to the RGB-JPEG format.

### 2.5. Machine Learning

Machine learning by AutoML was performed according to the website tutorials (https://cloud.google.com/vision/automl/docs/how-to). Firstly, we set up our projects and created a service account (https://cloud.google.com/vision/automl/docs/before-you-begin). A service account key, which is necessary to use AutoML, was created during this procedure. Images selected for machine learning were divided into a learning set and a test set. We prepared the images of the normal phenotype (control) and abnormal phenotypes (sorafenib-treated and wounded fishes for the angiogenesis and macrophage experiments, respectively) for each experiment. The numbers of images used in this study are summarized in Table 1. Then we created the datasets with a setting of "single-label classification (normal or abnormal)", imported each type of image file to the datasets, and labelled the imported images (https://cloud.google.com/vision/automl/docs/create-datasets). Finally, we trained AutoML, cloud-hosted the resulting models (https://cloud.google.com/vision/automl/docs/train), evaluated the models (https://cloud.google.com/vision/automl/docs/evaluate), and deployed them (https://cloud.google.com/vision/automl/docs/deploy). After completing the models, we retrieved the Project ID and Model ID from the "making individual predictions" page (https://cloud.google.com/vision/automl/docs/predict). These procedures are also explained on the ZF-ImageR website (https://github.com/YShimada0419/ZF-ImageR/wiki). The parameters, except dataset labelling (single-label classification), were set as default values. We trained AutoML once for each experiment.

**Table 1.** Numbers of zebrafish images for machine learning.

| Experiment | Normal Phenotypes | Abnormal Phenotypes |
|---|---|---|
| Angiogenesis [1] | 47 | 65 |
| Macrophage [2] | 104 | 156 |

[1] For angiogenesis-related experiment, the normal phenotypes were derived from the control group and the abnormal ones from the sorafenib (0.5 μM)-treated group. [2] For macrophage-related experiments, normal phenotypes were derived from the control group and abnormal ones from the wounded fishes.

### 2.6. ZF-ImageR

To upload the images in batches, we wrote the program ZF-ImageR using Python and then transferred it to an executable application software for Windows (.exe) and MacOS (.app). The Python code and the ZF-ImageR software can be downloaded from https://github.com/YShimada0419/ZF-

ImageR. See the Supplementary video (Video S1) for instructions on how to use. The Python code is shown and described as follows:

```
from google.cloud import automl_v1beta1 .... (1)
prediction_client = automl_v1beta1.PredictionServiceClient() .... (2)
prediction_client = prediction_client.from_service_account_json(KEY_FILE) .... (3)
name = 'projects/{}/locations/us-central1/models/{}'.format(project_id, model_id) .... (4)
payload = {'image': {'image_bytes': content}} .... (5)
request = prediction_client.predict(name, payload) .... (6)
print(request) .... (7)
```

(1)     Import Python library for Google Cloud.
(2)     Create Instance object of AutoML prediction service client.
(3)     Import Google Cloud service account key file which is required for accessing GCP services.
(4)     Set AutoML prediction model's ID which is required for (6).
(5)     Set Image data to post to AutoML for prediction which is required for (6).
(6)     Send a request to Google Cloud AutoML server for prediction of an image data from an service client which is prepared on (2).
(7)     Output prediction result.

## 3. Results

### 3.1. Evaluation of Zebrafish after Treatment with Anti-Angiogenesis Drug

Machine learning is used to build a mathematical model based on sample data, known as "training data", in order to make predictions for the "test data". We trained AutoML using multiple normal and abnormal zebrafish images (training data). Then, the newly prepared zebrafish images (test data) were used to diagnose them as normal or abnormal by the learned AutoML. The schematic representation of AutoML is depicted in Figure 1a. We first tested the learned AutoML using vascular-EGFP zebrafish with or without the treatment with the anti-angiogenic drug sorafenib. Sorafenib is an anticancer drug used for the treatment of several types of malignancies and it can inhibit multiple kinases such as vascular endothelial growth factor receptor, which suppresses tumor angiogenesis. As sorafenib suppresses vascular development in zebrafish embryos [14], we speculated that sorafenib-treated fishes would be suitable models for machine learning. According to a previous study, fishes treated with sorafenib (0.5 μM) from 24 hpf to 96 hpf could suppress the vascular development, and in particular the development of the intersegmental vessels (Figure 1b). AutoML was trained using 47 and 65 images of control (normal phenotype) and sorafenib-treated (abnormal phenotype) fishes, respectively. Then, the newly prepared images of the zebrafish with or without sorafenib were tested using the learned AutoML. Based on the results, the percentages "predicted as normal" and "predicted as abnormal" in the control group were 99.7 ± 0.2 and 0.31 ± 0.2%, respectively, while the percentages "predicted as normal" and "predicted as abnormal" in the sorafenib group were 6.0 ± 0.9% and 94.0 ± 0.9%, respectively (Figure 1c). This indicates that the learned AutoML was able to diagnose whether the input zebrafish image was from a control or sorafenib-treated sample. Further, we tested this learned AutoML (Figure 1c) with images of fishes treated with different concentrations (0–0.5 μM) of sorafenib (Figure 1d). As shown in Figure 1e, the prediction accuracy for samples "predicted as abnormal" (solid line) increased in a dose dependent manner and drastically increased between 0.25 and 0.5 μM. To compare the AutoML prediction and manual prediction, we asked three researchers to predict whether the zebrafish were normal or abnormal using the same images. For the prediction with or without 0.5 μM sorafenib (Figure 1b,c), the AutoML prediction was very similar to the manual prediction (Figure S1). In the dose-dependent experiment (Figure 1d,e), there was a similar tendency between the AutoML result and the manual prediction (Figure S2).

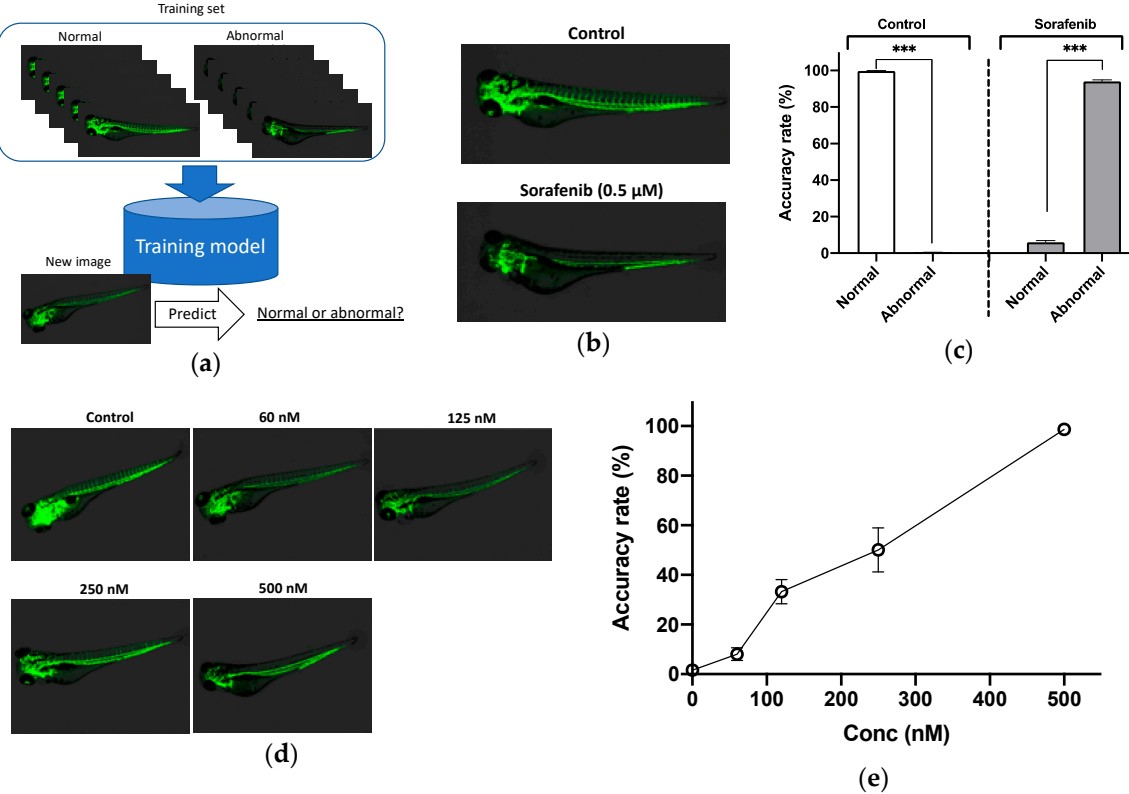

**Figure 1.** Machine learning to detect anti-angiogenic phenotype. (**a**) Schematic representation of the AutoML machine learning process. (**b**) Typical images of *Tg (kdrl:EGFP)* at 96 h-post fertilization (dpf) with or without sorafenib (0.5 μM). Sorafenib treatment was started at 24 hpf. The green color indicates vasculature. (**c**) Prediction of normal and abnormal phenotypes by the learned AutoML. The AutoML perfectly predicted the normal phenotype in the control group (left white bar) and abnormal phenotype in the sorafenib-treated group (right dark bars) with more than 90% accuracy. Note: *n* = 20, error bars indicate standard error (SE); *** *p* < 0.001. (**d**) Typical images of the test samples treated with different concentrations of sorafenib (0–500 nM). (**e**) Prediction of abnormal phenotypes of the sorafenib-treated fishes in a dose-dependent manner. The learned AutoML calculated the accuracy of the abnormal phenotype. Note: *n* = 15, error bars indicate SE.

## 3.2. Detection of Macrophage Abnormalities Using Machine Learning.

Next, we applied AutoML for other EGFP transgenic zebrafish, such as the macrophage-EGFP strain. It is well-known that macrophages can migrate to the wound site in the caudal fin as a part of the inflammatory response in zebrafish embryos [16]. We prepared the wounded larvae and uploaded their images into the learned AutoML (104 control and 156 wounded zebrafish). As shown in Figure 2a, macrophage accumulation (green spots) could be seen in the caudal fin. However, the learned AutoML could distinguish between these two phenotypes with less than 70% average accuracy in both the groups (Figure 2b). To improve the accuracy of AutoML prediction, we focused on the caudal fin by trimming the corresponding area (Figure 2c), but it still did not work (Figure 2d). Finally, we emphasized the green signals on the caudal fin (Figure 2e), and then the learned AutoML could effectively distinguish between normal and abnormal phenotype in the control and wounded group with more than 90% accuracy (Figure 2f). These results indicate that machine learning of image pre-processing is necessary for machine learning of macrophage-EGFP zebrafish. We also compared the final AutoML result (Figure 2f) with the manual prediction result, as shown in Figure S1, and found that these results are quite similar (Figure S3).

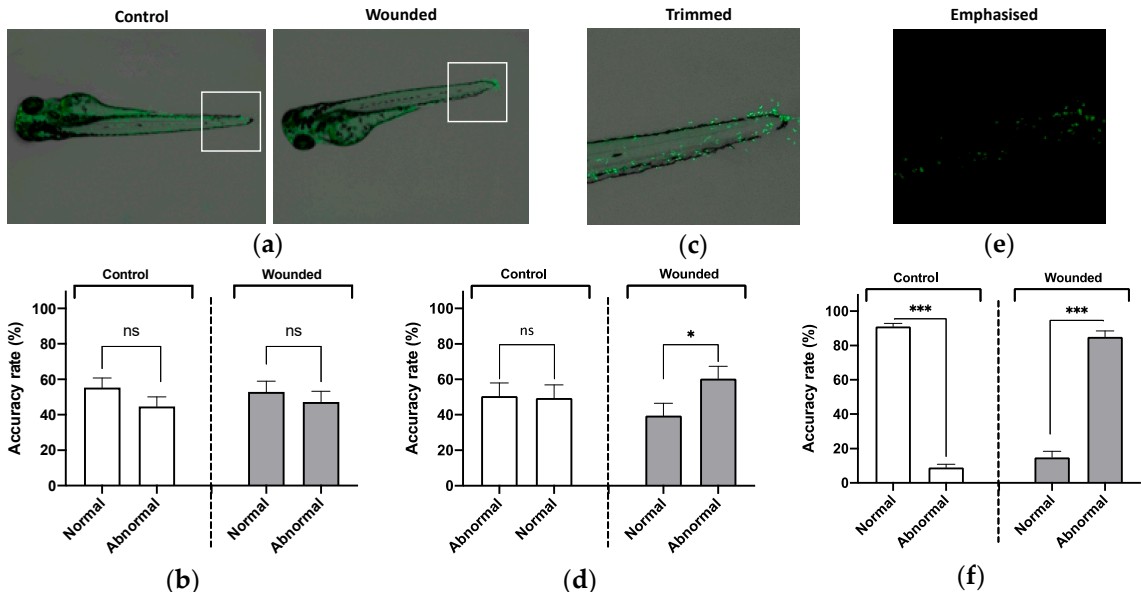

**Figure 2.** Machine learning to detect macrophage accumulation in the caudal fin. (**a**) Typical images of macrophage-Enhanced Green Fluorescent Protein (EGFP) zebrafish (72 hpf) with a wound in the caudal fin. (**b**) Prediction of normal and abnormal phenotypes by learned AutoML with an image set of (**a**). The prediction accuracies in each group were lower than 90%, indicating inappropriate machine learning. (**c**) Image pre-processing of caudal fin area (white square in (**a**)). Enlarged images of the caudal fin area. (**d**) Prediction of normal and abnormal phenotypes by learned AutoML with image from panel (**c**). The prediction accuracy in each group was still lower than 90%, indicating insufficient machine learning. (**e**) Image pre-processing of (**c**). Green-enhanced images of the caudal fin area. (**f**) Improved AutoML prediction (learned by image set of (**e**)) of macrophage accumulation in the caudal fin. Note: $n$ = 20, error bars indicate SE; * $p < 0.05$, *** $p < 0.001$.

## 4. Discussion

Zebrafish exhibit diverse and complex morphology, which makes them preferable over culture cells as a model for human disease. Moreover, it is possible to obtain high-throughput data equivalent to cells, which means that a large number of images can be processed in zebrafish screening experiments. It is almost impossible for researchers to visually analyze the images on such a large scale, and thus various analysis programs and software have been developed to achieve this. Despite various disadvantages, such as the cost of system introduction, difficulty in handling requiring a specialized programmer, and unsuitability for usage in a general laboratory, machine learning is one of the solutions to this problem. On the other hand, large IT companies offer machine learning or AI programs that do not require professional training. In particular, Google (AutoML) has released free web-based software for small scale applications, and these can be used for small scale development research of image recognition, such as face recognition.

Vascular-EGFP zebrafish are good samples for image acquisition, probably due to their fixed but complex structure. When we examined the images of the zebrafish treated with sorafenib using AutoML, small deviations in the accuracy rate were observed (Figure 1c). Although we could not deduce exactly why the prediction by AutoML was high, we speculate that it might be because of the focus (the area which is preferentially analyzed by the program) of AutoML, which was on the vascular structure (not the other parts of zebrafish such as the body). In other words, in the vascular-EGFP zebrafish study, the focus of AutoML is considered to be closer to the human eye because AutoML focuses on vasculature. However, in the macrophage-EGFP zebrafish study, the focus of AutoML is clearly different from that of the human eye. For example, in Figure 2a,b, AutoML, unlike human eyes, could not consciously observe the caudal fin area and was unable to focus, even after the area was enlarged (Figure 2c,d). Finally, by extracting only the EGFP (green signals) of the macrophages

from the magnified images (Figure 2e,f), AutoML could focus on the macrophages and identify those accumulated at the wound site. Machine learning can reveal the differences and discover algorithms from training images using a different perspective from that of humans. As seen in the case of our macrophage study, there could be other cases where it is not possible to obtain the results expected by humans. In such a case, prior image processing is required to support the machine learning. Alternatively, the researchers could take focused images in regions of interest or use only fluorescent images. Training AutoML with only fluorescent images would also improve prediction accuracy in vascular-EGFP zebrafish. To determine which experimental dataset needs image pre-processing, we used a trial-and-error approach in this study. Further trials using different types of zebrafish image datasets could determine the guidelines for image preparation, such as which magnifications to use or colors to emphasize in machine learning, especially for AutoML.

In this study, we utilized free machine learning programs for the diagnosis of zebrafish images. Ishaq et al. reported the deep learning-based classification of zebrafish deformation [11]. They used the neural network architecture AlexNet; neural networks and machine learning are different in principle. Neural networks are suitable for solving complex processes and require more thorough training compared to machine learning. Thus, for our purpose of determining which fish have anomalies, machine learning, especially supervised learning, is appropriate and simpler to use. Notably, small numbers of images are sufficient for training AutoML. For example, about 50 images were sufficient for our sorafenib experiment. In addition, it is known that repeated training on the same dataset can lead to different models that give slightly different results. However, there was no significant difference in prediction accuracy between the trainings with the same images in our vascular experiment (Figure S4).

While the free software is certainly attractive for many researchers, it has the disadvantage of being low-throughput. AutoML can upload training images in batches, but it requires uploading experimental images individually, which can take several minutes to upload 10 images. When combined with Google Cloud Storage, these test images can be uploaded in batches, although several command line commands are required to complete the task and it is not free (https://cloud.google.com/vision/automl/docs/predict-batch). To overcome this problem, we created a batch upload software "ZF-ImageR" specific for AutoML testing. As shown in the Supplementary video (Video S1), we can easily upload multiple images to the learned AutoML and retrieve the prediction for each image in a single CSV file at once. We have provided access to the Windows OS and Mac OS versions of the software as well as the source code on GitHub (https://github.com/YShimada0419/ZF-ImageR).

## 5. Conclusions

We succeeded in utilizing AutoML, a web-based machine learning program, to identify the abnormalities in the phenotypes of fluorescent zebrafish using their images. While few types of images need pre-processing to improve the focus for AutoML learning, the accuracy rates are similar to humans, and hence this program is applicable for high-throughput screening in combination with ZF-ImageR, a batch uploading software.

**Supplementary Materials:** The following are available online at http://www.mdpi.com/2411-5134/4/4/72/s1. Supplementary Methods: Image preprocessing procedure to emphasize green signals. Figure S1: Manual prediction of normal and abnormal phenotypes in zebrafish treated with 0.5 µM sorafenib. Figure S2: Manual prediction of normal and abnormal phenotypes of zebrafish treated with sorafenib in a dose-dependent manner. Figure S3: Manual prediction of normal and abnormal phenotypes in wounded zebrafish. Figure S4: Repeated training of AutoML with the same images did not affect the prediction accuracies in vascular-EGFP zebrafish. Video S1: How to use ZF-ImageR.

**Author Contributions:** Conceptualization, Y.S. and Y.N.; methodology, R.S. and H.N.; software, D.S.; validation, R.S. and D.S.; writing—original draft preparation, R.S.; writing—review and editing, Y.S.; project administration, Y.N.; funding acquisition, Y.S.

**Funding:** This research was funded by Japan Society for the Promotion of Science, grant number 17K08590.

**Acknowledgments:** The authors thank Rie Ikeyama for her secretarial assistance and Masako Inoue for breeding the zebrafish. We also thank Editage (www.editage.jp) for English language editing.

**Conflicts of Interest:** D.S. is an employee of RT Corporation, which is a robotics company. Y.N. is the CEO of RT Corporation. The other authors declare no conflict of interest directly relevant to the content of this article.

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
