# Peer review of "ZF-AutoML: An Easy Machine-Learning-Based Method to Detect Anomalies in Fluorescent-Labelled Zebrafish"

_inventions, doi:10.3390/inventions4040072_

Round 1
Reviewer 1 Report
General comments:
The manuscript “ZF-AutoML: An easy machine learning-based method to detect anomalies in fluorescent-labelled zebrafish” by Sawaki et al. describes the evaluation of cloud-based machine learning platforms (Google AutoML) for zebrafish image classification. These cloud platforms are targeted towards users with limited programming expertise or computing/programming resources. I appreciate that is a timely and fashionable topic, that is certainly of interest to the community, and that addresses a common challenge in automated microscopy in general and whole-organism screening in particular.
The manuscript can be considered a minor research note, reporting the evaluation of cloud-based machine learning methods for zebrafish screening using two small yet relevant benchmark experiments. Unfortunately, a direct application to larger datasets is not demonstrated, leaving the question unanswered, if this approach can be really used in screening assays. Moreover, the idea of using machine learning techniques for (zebrafish) screening data is not novel and has been demonstrated before. Nevertheless, the idea of using these cloud-platforms for this task is innovative and certainly of interest and idea-provoking to researchers.
However, the manuscript has major flaws and is partially poorly written. Certain sections lack the necessary details to reproduce the study. In particular, this heavily concerns the methods section and especially the pre-processing of data. Also, more experimental and validation data would be needed to support their claims. Moreover, to render the manuscript into a genuine protocol paper, I would suggest providing a step-by-step manual illustrating the practicalities of using AutoML. Furthermore, I have some issues with the drawn conclusion that I would like to see discussed (or addressed with additional data) in more detail.
Please find my detailed comments below.
Finally, I can only recommend this manuscript for publication in ‘inventions’ after substantial major revision.
Specific comments on sections:
Introduction:
The introduction would benefit by further details to guide the reader to the topic of ML/DL. I assume the main readerships of this manuscripts consists of zebrafish researchers or microscopists working on related topics, which are not familiar with the topic, and which search for new examples on how to approach challenging image analysis task in whole organism screening. It also needs better referencing to existing literature.
The authors describe other image analysis solutions by just highlighting two publications using either ImageJ or There are many more examples available in the literature including fully automated solutions for large-scale analysis of zebrafish screening, the latter currently being completely neglected by the authors. The authors should highlight and briefly discuss different analysis strategies for large-scale phenotyping. The authors should slightly elaborate on and include literature were ML/DL techniques have been used for whole image zebrafish phenotyping similar to their own approach (e.g. Ishaq et al), as the assumed readership will not be familiar with those techniques. I would suggest adding a few sentences to the introduction to explain what Machine/Deep Learning is in general, including its benefits is but also potential caveats.
Materials and Methods:
I have major issues with this section. It is currently insufficient and not acceptable. Several additions and modifications must be made to enable reproducibility, which in my opinion would be a prerequisite for considering this manuscript for publication.
The authors state that kdrl:EGFP were treated with SU-5416. However, this data is not shown anywhere in the manuscript. It would strengthen their claims if they included data on this compound too. What were the media conditions for compound exposure? Has it been pH buffered? Did they use DMSO? What served as control? I appreciate they reference another study in which this is presumably given, but it should be at least briefly mentioned here too. The authors must report the imaging conditions to enable users to reproduce their results: i.e. objective type and NA, exposure times used for BF and fluorescence. How were embryos mounted? In 96 well plates? Has embedding been used? The authors state that pre-pre-processing of image data is necessary, yet they only give a short list of software used for different steps. No details are given what was really done with these images, so the reader cannot reproduce what is reported here at all. This needs to be addressed. In detail: To what size the images were cropped to, how were images rotated, what angle and why was that done? I assume it was done to achieve a standardized orientation of fish? If the latter is true, this needs to be discussed in the Results and Discussion section, as then automated rotation of images would be required in large-scale assays, which is not a trivial automated image processing task for most wet-labs as it involves object recognition and analysis. The authors state GIMP was used ” to emphasize green colour”; it is totally unclear what that means? The authors must give more details and explain what kind of image alterations were carried out in order to use data for AutoML. After all these steps, was the resulting data then still a 14-bit Tiff or was it converted into some RGB-jpeg, I suppose the latter. This needs to be discussed. Just out of curiosity, is there any reason for using 3 different software packages for rather basic image manipulations? The authors report a GitHub repository for their uploading tool; however, the link is not functional and just gives a 404 error. Thus, I was not able to evaluate one of their core claims. However, based on the Suppl. Movie, I trust the authors that it will do the job, and they will make it publicly available upon publication as they have done with the ZFMapper. I urge the authors to make this tool available upon submission of the response letter. However, a short protocol on how to use that upload tool would be very helpful; it is not intuitive what e.g. the JSON file is needed for and where it comes from. Same is true for projectID, model ID, GCP Key file. This should be discussed in the paper or in the suggested protocol. No explanations about parameters or configuration of AutoML are given. As it is the core of the paper, the method section needs expansion on this matter.
Results:
The authors briefly describe the results of using AutoML for automatic classification of zebrafish data for data originating from 2 model experiments. Both experiments describe relevant zebrafish screening assays (vascular development and immune cell migration) and are well selected examples that are of interest to the zebrafish community. However, it would be required to provide a better evaluation of the model by using significantly larger sample sizes, and I have a few comments and questions that I would like to see addressed:
Can the author give more details on how AutoML was configured, or were the network training parameters left all just default (see also comments above)? The low amount of needed training data for the Sorafenib-Experiment is intriguing. I suggest the author include a statement stressing that this is a major advantage ,besides the obvious advantage of AutoML to be user-friendly and easy-to-use. The authors should include a statement of how often they trained the network until they got good results? Just once or were several attempts needed? It is known that a new training on the same dataset can lead to a different model that gives slightly different results. Was that evaluated? How is data trained in AutoML? Transfer learning or training from scratch? Any hints for the interested readers? (see also comments on Methods section above). Figure 1b,d: The authors show RGB overlay images which supposedly were also used for training. The author should comment on the possibility that the model just “learned” overall morphological changes such as bend tails and edemas. Can the authors provide data on models trained with fluorescence data only? This relates also to Figure2 and Discussion section, see comments below. Figure 1d-e: Can the authors generate data for models trained on embryos treated with 60nM, 125nM or 250 nM? Or maybe even a model trained with mixed data from all concentrations? That would be very interesting for the readers. Potentially, it would also overcome the apparent weakness of the current model to detect more subtle phenotypes that are easily scored manually. Figure 1e: As I understand the trained model was only tested on 5-7 embryos per group. This is not convincing at all. The author should perform experiments with larger batch sizes to validate the accuracy of the model and support their claim of having identified a method for large scale analysis. Also, the presented information in Fig1e is redundant, as ‘predicted as normal’ and ‘predicted as abnormal’ are just mirroring themself. Figure 2: I assume that the model was trained on data with both channels visible. So, it was also trained on overall morphology. As this is not drastically different between classes, this could be an explanation for the weak predictive power (in contrast to figure 1 where morphology could be the main driver for success). When only the fluorescence channel is used the results improve. The author state “Image pre-processing is necessary for machine learning of macrophage-EGFP zebrafish”. In my opinion, it could also be the opposite. I speculate that merging channels has caused the problem and one should train with data showing the relevant information only. All these observations and comparison of different data pre-processing steps are highly relevant for practical use of ML, so I do not suggest to change content, but I do not fully agree with the drawn conclusions. Also for the tail cut data, the sample sizes are very low (n=5), this should be significantly enhanced in a revised manuscript.
Discussion:
The authors should expand the discussion section with the points raised and explained above. The author speculate that AutoML was focusing on vasculature and Azure on overall morphology. I do not agree as the presented data suggest otherwise to me (please see above). If they have some evidence/hints for that, I suggest showing that in the manuscript. Are there any general guidelines for users how to pre-process data or do they suggest a try-and-error approach? What do they conclude how to prepare data, what are absolute pre-requisites? For example, would the ML/DL approach also work with non-rotated data. I do not fully understand the real benefit of the uploading tool. As I understand AutoML allows uploading zip-archives containing many images. Also in an online tutorial, I have seen that batch upload is possible. Can the authors elaborate on the rationale and benefit of this tool. A step-by-step protocol for using AutoML would be highly appreciated. This would help bench scientist testing the proposed approach (please also see above).
Minor comments:
English grammar should be checked and corrected
Author Response
Reviewer: 1
Thank you for giving us the opportunity to clarify some of the points made in our manuscript.
Major comments
Introduction:
The authors describe other image analysis solutions by just highlighting two publications using either ImageJ or There are many more examples available in the literature including fully automated solutions for large-scale analysis of zebrafish screening, the latter currently being completely neglected by the authors. The authors should highlight and briefly discuss different analysis strategies for large-scale phenotyping. The authors should slightly elaborate on and include literature were ML/DL techniques have been used for whole image zebrafish phenotyping similar to their own approach (e.g. Ishaq et al), as the assumed readership will not be familiar with those techniques.
Answer: We have described other automated solutions for zebrafish screening and discussed them briefly in the Introduction section. We also cited Ishaq et al. in the Introduction section and discussed their work in the Discussion section.
Page 1, line 39 (Introduction).
To evaluate the phenotypic changes, several types of image analysis procedures (software program, macros and these pipelines) have been developed and applied for large-scale zebrafish screening, such as ImageJ [1], custom R scripts [2], or commercial ones [3,4]. In general, freeware programs developed by researchers are semi-automatic and sometimes challenging to use, while commercial software programs for high-content images are easy to use, fully automatic, and expensive.
Page 7, line 264 (Discussion).
Ishaq et al. reported the deep learning-based classification of zebrafish deformation [11]. They used the neural network architecture AlexNet; neural network and machine learning are different in principle. Neural networks are suitable for solving complex processes and require more thorough training compared to machine learning. Thus, for our purpose of determining which fish have anomalies, machine learning, especially supervised learning, is appropriate and simpler to use. Notably, small numbers of images are sufficient for training AutoML. For example, about 50 images were sufficient for our sorafenib experiment. In addition, it is known that repeated training on the same dataset can lead to different models that give slightly different results. However, there was no significant difference in prediction accuracy between the trainings with the same images in our vascular experiment.
I would suggest adding a few sentences to the introduction to explain what Machine/Deep Learning is in general, including its benefits is but also potential caveats.
Answer: We have added these sentences in the Introduction, per your suggestion.
Page 2, line 56.
The combination of automatic imaging, quantification, and machine learning should promote large-scale, high-throughput zebrafish screening in the field of drug discovery. The attractiveness of machine learning is its superior efficacy, although the analysis pipeline is usually specific for a single purpose and cannot be easily generated, needing computer specialists and high cost. That means small-to-medium labs rarely use these systems.
Materials and Methods:
The authors state that kdrl:EGFP were treated with SU-5416. However, this data is not shown anywhere in the manuscript. It would strengthen their claims if they included data on this compound too. What were the media conditions for compound exposure? Has it been pH buffered? Did they use DMSO? What served as control? I appreciate they reference another study in which this is presumably given, but it should be at least briefly mentioned here too.
Answer: We are sorry for the confusion. We did not use SU-5416. We have removed it from our manuscript.
The authors must report the imaging conditions to enable users to reproduce their results: i.e. objective type and NA, exposure times used for BF and fluorescence. How were embryos mounted? In 96 well plates? Has embedding been used?
Answer: We have added the requested details to the Materials and Methods section.
Page 2, line 86 (Materials and Methods).
For imaging, zebrafish were anaesthetised with 0.003% tricaine (MS222; Sigma-Aldrich, St. Louis, MO, USA) and mounted laterally on 10-cm dishes coated with 1% agarose dissolved in H2O. The conditions for image capture were as follows:
4× objective lens with 10× eyepieces (40× total magnification).
High-resolution mode (1920 × 1440 px).
Exposure time; brightfield: 1/7500 s, green fluorescent protein
(GFP: Ex 470/40, Em 525/50): 1.2 s.
The orientation of zebrafish was random.
Merged images (automatically prepared using the BZ-H3AE software; Keyence) were used for this study.
The authors state that pre-pre-processing of image data is necessary, yet they only give a short list of software used for different steps. No details are given what was really done with these images, so the reader cannot reproduce what is reported here at all. This needs to be addressed. In detail: To what size the images were cropped to, how were images rotated, what angle and why was that done? I assume it was done to achieve a standardized orientation of fish? If the latter is true, this needs to be discussed in the Results and Discussion section, as then automated rotation of images would be required in large-scale assays, which is not a trivial automated image processing task for most wet-labs as it involves object recognition and analysis.
Answer: We added a new subsection “2.4. Image processing” in the Materials and Methods section and the supplementary method to explain this procedure in detail. In addition, for machine learning, we did not standardize the orientation of the fish.
Page 3, line 96 (Materials and Methods).
2.4. Image processing
Images of the macrophage assay were pre-processed for machine learning. Microsoft Paint was used to partially cut the image including the wound area (size: 631 × 601 px). ImageMagick (Version 7.0; ImageMagick Studio, Landenberg, PA, USA) was used to rotate the images. ImageMagick and GIMP software (version 2.10; C Orinda, CA, USA) were used to emphasize the green colour of the images. For a step-by-step protocol, please see the supplementary method. In addition, to reduce the file size and upload time of images to AutoML, we converted TIFF files to the RGB-JPEG format.
Page 6, line 246 (Discussion).
In addition, there is no difference in the learning efficacy of AutoML between random- and standardized-orientation (left-headed) fish images.
The authors state GIMP was used ” to emphasize green colour”; it is totally unclear what that means? The authors must give more details and explain what kind of image alterations were carried out in order to use data for AutoML. After all these steps, was the resulting data then still a 14-bit Tiff or was it converted into some RGB-jpeg, I suppose the latter.
Answer: We have added a supplementary method describing how to use GIMP in detail. Also, we converted 14-bit TIFF to RGB-JPEG to shorten the uploading time for zebrafish images.
Page 3, line 101 (Materials and Methods).
In addition, to reduce the file size and upload time of images to AutoML, we converted TIFF files to the RGB-JPEG format.
Just out of curiosity, is there any reason for using 3 different software packages for rather basic image manipulations? The authors report a GitHub repository for their uploading tool; however, the link is not functional and just gives a 404 error. Thus, I was not able to evaluate one of their core claims. However, based on the Suppl. Movie, I trust the authors that it will do the job, and they will make it publicly available upon publication as they have done with the ZFMapper. I urge the authors to make this tool available upon submission of the response letter.
Answer: We planned to upload ZF-ImageR to GitHub after our manuscript was approved, but we have now uploaded it according to your request.
However, a short protocol on how to use that upload tool would be very helpful; it is not intuitive what e.g. the JSON file is needed for and where it comes from. The same is true for projectID, model ID, GCP Key file. This should be discussed in the paper or in the suggested protocol.
Answer: We have added the requested details to the Materials and Methods section.
Page 3, line 105 (Materials and Methods).
Machine learning by AutoML was performed according to the website tutorials (https://cloud.google.com/vision/automl/docs/how-to). Firstly, we set up our projects and created a service account (https://cloud.google.com/vision/automl/docs/before-you-begin). A service account key, which is necessary to use AutoML, was created during this procedure. Images selected for machine learning were divided into a learning set and a test set. We prepared the images of the normal phenotype (control) and abnormal phenotypes (sorafenib-treated and wounded fishes for the angiogenesis and macrophage experiments, respectively) for each experiment. The numbers of images used in this study are summarized in Table 1. Then we created the datasets with a setting of ‘Single-label classification (normal or abnormal)’, imported each type of image file to the datasets, and labelled the imported images (https://cloud.google.com/vision/automl/docs/create-datasets). Finally, we trained AutoML, cloud-hosted the resulting models (https://cloud.google.com/vision/automl/docs/train), evaluated the models (https://cloud.google.com/vision/automl/docs/evaluate), and deployed them (https://cloud.google.com/vision/automl/docs/deploy). After completing the models, we retrieved the Project ID and Model ID from the ‘Making individual predictions’ page (https://cloud.google.com/vision/automl/docs/predict). These procedures are also explained on the ZF-ImageR website (https://github.com/YShimada0419/ZF-ImageR/wiki).
No explanations about parameters or configuration of AutoML are given. As it is the core of the paper, the method section needs expansion on this matter.
Answer: We only chose “Single-label classification” for creation of the datasets. Other parameters such as confidence and threshold were set as default values.
Page 3, line 112 (Materials and Methods).
Then we created the datasets with a setting of ‘Single-label classification (normal or abnormal)’,
Page 3, line 121 (Materials and Methods).
The parameters, except dataset labelling (single-label classification), were set as default values.
Results:
Can the author give more details on how AutoML was configured, or were the network training parameters left all just default (see also comments above)?
Answer: We only chose “Single-label classification” for creation of the datasets. Other parameters such as confidence and threshold were set as default values.
Page 3, line 112 (Materials and Methods).
Then we created the datasets with a setting of ‘Single-label classification (normal or abnormal)’,
Page 3, line 121 (Materials and Methods).
The parameters, except dataset labelling (single-label classification), were set as default values.
The low amount of needed training data for the Sorafenib-Experiment is intriguing. I suggest the author include a statement stressing that this is a major advantage, besides the obvious advantage of AutoML to be user-friendly and easy-to-use.
Answer: I have added the statement to the Discussion.
Page 7, line 269.
Notably, small numbers of images are sufficient for training AutoML. For example, about 50 images were sufficient for our sorafenib experiment.
The authors should include a statement of how often they trained the network until they got good results? Just once or were several attempts needed?
Answer: We trained the network only once to obtain the results in this paper. We do not think it needs multiple attempts to have good (or bad in the case of the macrophage images) results in AutoML. We have added this statement to the Materials and Methods section.
Page 3, line 122 (Materials and Methods).
We trained AutoML once for each experiment.
It is known that a new training on the same dataset can lead to a different model that gives slightly different results. Was that evaluated?
Answer: Yes, but in our case, there was no significant difference in prediction accuracy between the trainings with the same images.
Page 7, line 270 (Discussion).
In addition, it is known that repeated training on the same dataset can lead to different models that give slightly different results. However, there was no significant difference in prediction accuracy between the trainings with the same images in our vascular experiment.
How is data trained in AutoML? Transfer learning or training from scratch? Any hints for the interested readers? (see also comments on Methods section above).
Answer: We have answered your question in “2.5. Machine learning” in the Materials and Methods.
Page 3, line 105 (Materials and Methods).
Machine learning by AutoML was performed according to the website tutorials (https://cloud.google.com/vision/automl/docs/how-to). Firstly, we set up our projects and created a service account (https://cloud.google.com/vision/automl/docs/before-you-begin). A service account key, which is necessary to use AutoML, was created during this procedure. Images selected for machine learning were divided into a learning set and a test set. We prepared the images of the normal phenotype (control) and abnormal phenotypes (sorafenib-treated and wounded fishes for the angiogenesis and macrophage experiments, respectively) for each experiment. The numbers of images used in this study are summarized in Table 1. Then we created the datasets with a setting of ‘Single-label classification (normal or abnormal)’, imported each type of image file to the datasets, and labelled the imported images (https://cloud.google.com/vision/automl/docs/create-datasets). Finally, we trained AutoML, cloud-hosted the resulting models (https://cloud.google.com/vision/automl/docs/train), evaluated the models (https://cloud.google.com/vision/automl/docs/evaluate), and deployed them (https://cloud.google.com/vision/automl/docs/deploy). After completing the models, we retrieved the Project ID and Model ID from the ‘Making individual predictions’ page (https://cloud.google.com/vision/automl/docs/predict). These procedures are also explained on the ZF-ImageR website (https://github.com/YShimada0419/ZF-ImageR/wiki). The parameters, except dataset labelling (single-label classification), were set as default values. We trained AutoML once for each experiment.
Figure 1b,d: The authors show RGB overlay images which supposedly were also used for training. The author should comment on the possibility that the model just “learned” overall morphological changes such as bend tails and edemas. Can the authors provide data on models trained with fluorescence data only? This relates also to Figure2 and Discussion section, see comments below.
Answer: We are sure that training with only fluorescent images improves prediction accuracy. However, even when using the merged images, the trained AutoML can detect abnormalities in fish caused by angiogenesis inhibitor sorafenib. We would like to emphasize the power of AutoML in the vascular zebrafish experiment in Figure 1, which was not enough for macrophage experiment in Figure 2. Unfortunately, we do not have enough time for completing new training for your request in the limited time for the revision.
Page 7, line 257 (Discussion).
Training AutoML with only fluorescent images would also improve prediction accuracy in vascular-EGFP zebrafish.
Figure 1d-e: Can the authors generate data for models trained on embryos treated with 60nM, 125nM or 250 nM? Or maybe even a model trained with mixed data from all concentrations? That would be very interesting for the readers. That would be very interesting for the readers. Potentially, it would also overcome the apparent weakness of the current model to detect more subtle phenotypes that are easily scored manually.
Answer: We applied the learned AutoML in Fig. 1c to the sorafenib dose-dependent experiment (Fig. 1d-e).
Page 4, line 178 (Results).
Further, we tested this learned AutoML (Fig. 1c) with images of fishes treated with different concentrations (0 – 0.5 μM) of sorafenib (Fig. 1d).
Figure 1e: As I understand the trained model was only tested on 5-7 embryos per group. This is not convincing at all. The author should perform experiments with larger batch sizes to validate the accuracy of the model and support their claim of having identified a method for large scale analysis. Also, the presented information in Fig1e is redundant, as ‘predicted as normal’ and ‘predicted as abnormal’ are just mirroring themselves.
Answer: We performed the experiments again to increase the number of test images and found that “predicted as normal” and “predicted as abnormal” are indeed mirroring, so we removed “predicted as normal”.
Page 5, line 197 (Figure 1).
The learned AutoML calculated accuracy of abnormal phenotype. n = 15, error bars indicate SE.
Figure 2: I assume that the model was trained on data with both channels visible. So, it was also trained on overall morphology. As this is not drastically different between classes, this could be an explanation for the weak predictive power (in contrast to figure 1 where morphology could be the main driver for success). When only the fluorescence channel is used the results improve. The author state “Image pre-processing is necessary for machine learning of macrophage-EGFP zebrafish”. In my opinion, it could also be the opposite. I speculate that merging channels has caused the problem and one should train with data showing the relevant information only. All these observations and comparison of different data pre-processing steps are highly relevant for practical use of ML, so I do not suggest to change content, but I do not fully agree with the drawn conclusions.
Answer: We agree that using merged images would weaken the power of machine learning. However, in the biology field, we usually use merged images (bright-field with fluorescent observations) for representative figures, so we first tried to use merged images in the vascular zebrafish experiments. The merged images are “post-processed” images and the fluorescent images are “not processed” images.
Page 7, line 254 (Discussion).
As seen in the case of our macrophage study, there can be other cases where it is not possible to obtain the results expected by humans. In such a case, prior image processing is required to support the machine learning. Alternatively, the researchers could take focused images in regions of interest or use only fluorescent images.
Also for the tail cut data, the sample sizes are very low (n=5), this should be significantly enhanced in a revised manuscript.
Answer: We have increased the numbers of test images to n = 20.
Page 6, line 223 (Figure 2).
(f) Improved AutoML prediction (learned by image set of e) of macrophage accumulation in the caudal fin. n = 20, error bars indicate SE. ***p < 0.001.
Discussion:
The authors should expand the discussion section with the points raised and explained above. The author speculate that AutoML was focusing on vasculature and Azure on overall morphology. I do not agree as the presented data suggest otherwise to me (please see above). If they have some evidence/hints for that, I suggest showing that in the manuscript.
Answer: We have added the statement below.
Page 6, line 245.
AutoML focuses on vasculature, while Azure focuses on overall morphology.
Are there any general guidelines for users how to pre-process data or do they suggest a try-and-error approach? What do they conclude how to prepare data, what are absolute pre-requisites? For example, would the ML/DL approach also work with non-rotated data.
Answer: From our limited study, it is hard to predict which machine learning datasets require pre-processing and the type of processing without using a trial-and-error approach. It is our impression that the learning ability of AutoML is similar to that of our new undergraduate students. They know little about the ROI in the zebrafish images in their experiments, and they need training from their supervisor. However, this is just our impression and is not suitable for inclusion in our manuscript. Also, we do not need standardized oriented fish images for AutoML.
Page 7, line 258.
To determine which experimental dataset needs image pre-processing, we used a trial-and-error approach in this study. Further trials using different types of zebrafish image datasets could determine the guidelines for image preparation, such as which magnifications to use or colours to emphasize in machine learning, especially for AutoML.
I do not fully understand the real benefit of the uploading tool. As I understand AutoML allows uploading zip-archives containing many images. Also, in an online tutorial, I have seen that batch upload is possible. Can the authors elaborate on the rationale and benefit of this tool?
Answer: We can easily upload images for training in a batch manner, but not for the tests. But we now know that we can also upload test images in batches by using Google Cloud Storage with command line (https://cloud.google.com/vision/automl/docs/predict-batch). Our ZF-ImageR can also upload images in batches, get the prediction result for each image, and create a single CSV file for the batch. We do not have to use Google Cloud Storage to use AutoML, which means it is completely free. However, ZF-ImageR is easier than using command line commands with Google Cloud Storage.
Page 7, line 275.
AutoML can upload training images in batches, but it requires uploading experimental images individually, which can take several minutes to upload 10 images. When combined with Google Cloud Storage, these test images can be uploaded in batches, although several command line commands are required to complete the task and it is not free (https://cloud.google.com/vision/automl/docs/predict-batch). To overcome this problem, we created a batch upload software “ZF-ImageR” specific for AutoML test. As shown in the supplementary movie (Video S1), we can easily upload multiple images to the learned AutoML and retrieve the prediction for each image in a single CSV file at once.
A step-by-step protocol for using AutoML would be highly appreciated. This would help bench scientist testing the proposed approach (please also see above).
Answer: We have included the requested protocol in the Materials and Methods section (2.5. Machine learning).
Page 3, line 105 (Materials and Methods).
Machine learning by AutoML was performed according to the website tutorials (https://cloud.google.com/vision/automl/docs/how-to). Firstly, we set up our projects and created a service account (https://cloud.google.com/vision/automl/docs/before-you-begin). A service account key, which is necessary to use AutoML, was created during this procedure. Images selected for machine learning were divided into a learning set and a test set. We prepared the images of the normal phenotype (control) and abnormal phenotypes (sorafenib-treated and wounded fishes for the angiogenesis and macrophage experiments, respectively) for each experiment. The numbers of images used in this study are summarized in Table 1. Then we created the datasets with a setting of ‘Single-label classification (normal or abnormal)’, imported each type of image file to the datasets, and labelled the imported images (https://cloud.google.com/vision/automl/docs/create-datasets). Finally, we trained AutoML, cloud-hosted the resulting models (https://cloud.google.com/vision/automl/docs/train), evaluated the models (https://cloud.google.com/vision/automl/docs/evaluate), and deployed them (https://cloud.google.com/vision/automl/docs/deploy). After completing the models, we retrieved the Project ID and Model ID from the ‘Making individual predictions’ page (https://cloud.google.com/vision/automl/docs/predict). These procedures are also explained on the ZF-ImageR website (https://github.com/YShimada0419/ZF-ImageR/wiki). The parameters, except dataset labelling (single-label classification), were set as default values. We trained AutoML once for each experiment.
Minor comments:
English grammar should be checked and corrected
Answer: We have hired the professional English proofreading company, Editage.
Reviewer 2 Report
This is an interesting study which combined the approaches of machine learning and fluorescent imaging in the phenotypic study of zebrafish model. The authors were able to show the power of their software in automatically processing and analyzing the figures. I only have two concerns:
For both experiments, the authors only showed the results of machine learning. However, in order to demonstrate the validity of these predictive results, it is necessary to compare the predictive accuracy by machine versus those predicted manually, and maybe, to further examine why some figures, if any, are inconsistently judged by machine and human beings, to better clarify the point and provide more information.
Similarly, for the dose-dependent study of Sorafenib, the authors only showed the corresponding changes of abnormality rates predicted by AutoML (Figure 1e), which did make sense. However, it is still important to firstly demonstrate that Sorafenib did dose-dependently inhibit zebrafish angiogenesis in the dose range tested in the study, maybe through manually analysis.
Author Response
Reviewer: 2
Thank you for giving us the opportunity to clarify some of the points made in our manuscript.
Major comments
For both experiments, the authors only showed the results of machine learning. However, in order to demonstrate the validity of these predictive results, it is necessary to compare the predictive accuracy by machine versus those predicted manually, and maybe, to further examine why some figures, if any, are inconsistently judged by machine and human beings, to better clarify the point and provide more information.
Answer: According to your suggestion, we have asked three researchers to perform manual predictions for the experiments associated with Figures 1 and 2 and compared their results with those of the machine learning in the supplementary figures.
Page 5, line 181.
To compare the AutoML prediction and manual prediction, we asked three researchers to predict whether the zebrafish were normal or abnormal using the same images. For the prediction with or without 0.5 µM sorafenib (Fig. 1b and c), the AutoML prediction was very similar to the manual prediction (Fig. S1). In the dose-dependent experiment (Fig. 1d and e), there was a similar tendency between the AutoML result and the manual prediction (Fig. S2).
Page 6, line 211.
We also compared the final AutoML result (Fig. 2f) with the manual prediction result as shown in Fig. S1 and found that these results are quite similar (Fig. S3).
Similarly, for the dose-dependent study of Sorafenib, the authors only showed the corresponding changes of abnormality rates predicted by AutoML (Figure 1e), which did make sense. However, it is still important to firstly demonstrate that Sorafenib did dose-dependently inhibit zebrafish angiogenesis in the dose range tested in the study, maybe through manually analysis.
Answer: We hope our new results for manual analysis for Figure 2e as supplementary Figure S2.
Page 5, line 184.
In the dose-dependent experiment (Fig. 1d and e), there was a similar tendency between the AutoML result and the manual prediction (Fig. S2).
Round 2
Reviewer 1 Report
I would like to thank the author for their response. The authors have addressed most of my comments and I think the manuscript has improved. However, there are major points that must be addressed before this manuscript can be considered for publication. Moreover, there are still several comments and suggestions, that I would like to see addressed in a further revision.
Major points:
The authors report now a higher sample size N=20 instead of N=5. However, they still report the same accuracies for the Sorafenib experiments as in the initial version (line 168, bar charts in Fig1c were not updated in the revised version). How can that be? Have they simply taken the same data 4 times or was the manuscript not updated properly? Both would be inacceptable and must be resolved. I did not spot it in the first version, but the labelling of Fig2 (a,b,c,d,e,f) seems incorrect, i.e. it is not matching entirely the figure legend and the text.Minor points:
The authors state now in Line 42 “In general, freeware programs developed by researchers are semi-automatic and sometimes challenging to use, while commercial software programs for high-content images are easy to use, fully automatic, and expensive.”
I understand the intention of this statement. However, I would argue that there are also fully automated analysis solutions for large scale phenotyping, which are often very application specific and indeed challenging to use for zebrafish image analysis. On the other hand, there are open-source solutions for cell-based assays that are powerful and relatively easy-to-use (e.g. Cellprofiler). Similarly, I am not aware of an easy-to-use and generic commercial software for zebrafish screening, at least not for the e.g. scoring of vessel sprouting or cell counting after tail cut. I suggest that the authors rephrase their current statement, as I think it is too general and not entirely valid.
The author now cite Ishaq et al and discuss it. I want to say that I am not involved in this work at all, but think it is a relevant addition that’s why I suggested it as an example. They discuss that it is based on Deep Learning and AutoML on machine learning, which is different according to the authors. Are the author confident that AutoML is not relying on some Deep Learning techniques too? In summary, I would like to see a brief statement in the manuscript of how AutoML is in principle working if this information is available from Google, and adjust the statement in line 264ff. Just saying supervised machine learning is different to deep learning is potentially too vague.
The author should expand the description of the Imaging conditions by the numerical aperture of the objective used.
The author state now that exposure time for brightfield is 1/7500s which corresponds to 0.13 milliseconds. That sounds strangely low. Similarly, the reported exposure time for fluorescence seems rather high.
In Line246: When there is no difference in learning efficiency, why the author have rotated the images? If there is no difference in learning efficiency (as they state), I assume the authors have data on this. This could be reported in Supplementary Info. If it was not evaluated then it should be said that only standard orientation was explored.
In line 122 the author state they trained only once, and in line 270 they state that repeated training did not lead to significant differences. That is contradicting and needs to be resolved.
The author state that they answered my question on how AutoML is learning in section 2.5. I do not agree, that is a description of the practical procedure, not a statement on the background of AutoML.
In Line 245 author state: “AutoML focuses on vasculature, while Azure focuses on overall morphology.”
As the authors also say, this is pure speculation and there is absolutely no evidence provided that would underscore this statement. As a general comment, I think the comparison to Azure is actually not adding much to the paper and is slightly confusing to readers. It is too broad, lacks scientific or technical evidence, and it is merely observational and speculative. To clarifiy my point, I think overall the manuscript is written from a biologist and non-expert point of view that is using AutoML and machine learning for analysis, while seeing it as a “black box” that is somehow doing the job. This is fine as a lot of users would maybe follow a similar approach (if good or bad is a different discussion). I also appreciate the paper in general, as it is idea provoking to other bench scientist. However, making comparative statements about the usage and analytical power of Microsoft Azure vs Google AutoML feels beyond the scope of the manuscript and is not adding anything. It rather underlines the somewhat naïve approach of the authors. However, I leave it to the authors if they want to keep that comparison.
Author Response
Thank you for giving us the opportunity to clarify some of the points made in our manuscript.
Major points:
The authors report now a higher sample size N=20 instead of N=5. However, they still report the same accuracies for the Sorafenib experiments as in the initial version (line 168, bar charts in Fig1c were not updated in the revised version). How can that be? Have they simply taken the same data 4 times or was the manuscript not updated properly? Both would be inacceptable and must be resolved.
Answer: We are sorry for our mistake not updating the main text. We have corrected them. And of course, we used new images for the analysis, not using the same images 4 times. We also made the same mistakes for Figure 1e in the main text and the figure legends.
Page 4, line 166.
Based on the results, the percentage of “predicted as normal” and “predicted as abnormal” in the control group was 99.7 ±0.2 and 0.31 ±0.2%, respectively, while the percentage of “predicted as normal” and “predicted as abnormal” in the sorafenib group was 6.0 ±0.9% and 94.0 ±0.9%, respectively (Fig. 1c).
Page 4, line 172.
As shown in Figure 1e, the prediction accuracy for samples “predicted as abnormal” (solid line) increased in a dose dependent manner and drastically increased between 0.25 – 0.5 µM.
Page 5, line 187.
Prediction of abnormal phenotypes of the sorafenib-treated fishes in a dose-dependent manner.
I did not spot it in the first version, but the labelling of Fig2 (a,b,c,d,e,f) seems incorrect, i.e. it is not matching entirely the figure legend and the text.
Answer: We are sorry for our careless mistakes. We have rearranged the labelling of Figure 2.
Minor points:
In Line246: When there is no difference in learning efficiency, why the authors have rotated the images? If there is no difference in learning efficiency (as they state), I assume the authors have data on this. This could be reported in Supplementary Info. If it was not evaluated then it should be said that only standard orientation was explored.
Answer: In this study, we did not rotate images during machine learning and their testing, that means we just input random-oriented fish images to AutoML. For figures, it is usual that fish is oriented left-headed, so we just choose (or rotate) them for figure images in our manuscript. We have removed this statement.
In line 122 the author state they trained only once, and in line 270 they state that repeated training did not lead to significant differences. That is contradicting and needs to be resolved.
Answer: We performed another learning for the last revision but seems contradicting as you comment. For the Fig. 1 and 2, we performed one-time (1st trial) learned AutoML. We added the comparison data between first and second trial as supplementary figure (Fig. S4).
Page 7, line 259.
However, there was no significant difference in prediction accuracy between the trainings with the same images in our vascular experiment (Fig. S4).
The author state that they answered my question on how AutoML is learning in section 2.5. I do not agree, that is a description of the practical procedure, not a statement on the background of AutoML.
Answer: I suppose you would like to how AutoML works. We have searched intensively the principle of AutoML’s algorism but cannot find it out. We conclude that the AutoML is a kind of black box, as you mentioned below. That is why we just state the description of the practical procedure only.
In Line 245 author state: “AutoML focuses on vasculature, while Azure focuses on overall morphology.”
As the authors also say, this is pure speculation and there is absolutely no evidence provided that would underscore this statement. As a general comment, I think the comparison to Azure is actually not adding much to the paper and is slightly confusing to readers. It is too broad, lacks scientific or technical evidence, and it is merely observational and speculative. To clarify my point, I think overall the manuscript is written from a biologist and non-expert point of view that is using AutoML and machine learning for analysis, while seeing it as a “black box” that is somehow doing the job. This is fine as a lot of users would maybe follow a similar approach (if good or bad is a different discussion). I also appreciate the paper in general, as it is idea provoking to other bench scientist. However, making comparative statements about the usage and analytical power of Microsoft Azure vs Google AutoML feels beyond the scope of the manuscript and is not adding anything. It rather underlines the somewhat naïve approach of the authors. However, I leave it to the authors if they want to keep that comparison.
Answer: Thank you very much for your discussion. We completely agree your opinion. In addition, because the algorisms of these machine learning are updating every month, so we expect that Azure will also be useful for zebrafish imaging near future. Thus, we have removed the statements about Azure from Results and Discussions.
Round 3
Reviewer 1 Report
Thank you for the constructive and responsive review process. I suggest to accept the paper in its current form.